# Allogenic Stem Cells Carried by Porous Silicon Scaffolds for Active Bone Regeneration In Vivo

**DOI:** 10.3390/bioengineering10070852

**Published:** 2023-07-19

**Authors:** Matthieu Renaud, Philippe Bousquet, Gerard Macias, Gael Y. Rochefort, Jean-Olivier Durand, Lluis F. Marsal, Frédéric Cuisinier, Frédérique Cunin, Pierre-Yves Collart-Dutilleul

**Affiliations:** 1Laboratoire Biosanté et Nanoscience (LBN), Université Montpellier, 34000 Montpellier, France; matt.renaud18@live.fr (M.R.); frederic.cuisinier@umontpellier.fr (F.C.); 2Faculty of Dentistry, Université de Tours, 37000 Tours, France; 3Faculty of Dentistry, Université Montpellier, 34000 Montpellier, France; 4Service Odontologie, Hospital Center University de Montpellier, 34000 Montpellier, France; 5Institute Charles Gerhardt Montpellier (ICGM), Université Montpellier, Centre National de la Recherche Scientifique (CNRS), ENSCM, 34000 Montpellier, France; 6Department of Electronic, Electrical and Automatic Engineering (DEEEA), Universitat Rovira i Virgili, 43003 Tarragona, Spain

**Keywords:** tissue engineering, bone regeneration, mesenchymal stem cells, bone substitute, dental pulp stem cells, porous silicon

## Abstract

To date, bone regeneration techniques use many biomaterials for bone grafting with limited efficiencies. For this purpose, tissue engineering combining biomaterials and stem cells is an important avenue of development to improve bone regeneration. Among potentially usable non-toxic and bioresorbable scaffolds, porous silicon (pSi) is an interesting biomaterial for bone engineering. The possibility of modifying its surface can allow a better cellular adhesion as well as a control of its rate of resorption. Moreover, release of silicic acid upon resorption of its nanostructure has been previously proved to enhance stem cell osteodifferentiation by inducing calcium phosphate formation. In the present study, we used a rat tail model to experiment bone tissue engineering with a critical size defect. Two groups with five rats per group of male Wistar rats were used. In each rat, four vertebrae were used for biomaterial implantation. Randomized bone defects were filled with pSi particles alone or pSi particles carrying dental pulp stem cells (DPSC). Regeneration was evaluated in comparison to empty defect and defects filled with xenogenic bone substitute (Bio-Oss^®^). Fluorescence microscopy and SEM evaluations showed adhesion of DPSCs on pSi particles with cells exhibiting distribution throughout the biomaterial. Histological analyzes revealed the formation of a collagen network when the defects were filled with pSi, unlike the positive control using Bio-Oss^®^. Overall bone formation was objectivated with µCT analysis and showed a higher bone mineral density with pSi particles combining DPSC. Immunohistochemical assays confirmed the increased expression of bone markers (osteocalcin) when pSi particles carried DPSC. Surprisingly, no grafted cells remained in the regenerated area after one month of healing, even though the grafting of DPSC clearly increased bone regeneration for both bone marker expression and overall bone formation objectivated with µCT. In conclusion, our results show that the association of pSi with DPSCs in vivo leads to greater bone formation, compared to a pSi graft without DPSCs. Our results highlight the paracrine role of grafted stem cells by recruitment and stimulation of endogenous cells.

## 1. Introduction

Reconstruction of large maxillar and mandibular bony defects is a frequent surgical challenge, especially after tooth extraction, traumas, diseases, or surgical procedures. During implant therapy, lack of sufficient alveolar bone leads to esthetical and functional problems with social and health impacts [1].

Many procedures have been proposed to treat alveolar defects, using autograft, allograft or alveolar preservation [2,3,4,5]. Unfortunately, these therapeutical approaches are complex and did not succeed in achieving ideal bone reconstruction. Furthermore, allograft procedures have shown limitations such as disease transmission, immunogenic response, and nonunion [6].

A tissue engineering approach has been developed using various bioactive factors, such as bone morphogenic protein (BMP), amelogenin or autologous platelet concentrates with interesting enhancement of bone regeneration processes and the reduction of postoperative discomfort, but do not have sufficient evidence of a significant effect on hard tissue regeneration [7]. Optimal carriers for bioactive agents are still to be investigated, considering the timing of administration combination of different agents for synergistic effects. This tissue engineering approach can also include the use of progenitor cells, brought to the injured site. Thus, for bone defect repair, combining osteogenic capacities of cells with appropriate scaffolding material carrying cells to the injured bone site represents promising therapeutics [8]. The ideal biomaterial scaffold should support cell attachment, proliferation and differentiation. Scaffolds must be designed to induce bone formation and vascularization. Thus, they are often porous, and made of bioresorbable materials harboring different growth factors, drugs, or stem cells [9,10,11,12].

Therefore, we aimed to investigate these various aspects of progenitor cells delivery and scaffold design. We used a model of mesenchymal stem cells coupled with a silicon-based resorbable biomaterial. Dental pulp stem cells (DPSC) are mesenchymal stem cells embryologically derived from neural crest, and migrating into mesenchyme. They are capable of several differentiation pathways, including osteoblasts when cultured in an appropriate culture medium, and can form a complete and well-vascularized lamellar bone in vivo [1]. DPSC are capable of long-term cultivation without changing their viability, phenotype, or genotype [13].

Porous silicon (pSi) is a bioresorbable structure that has been extensively studied in various areas of the biomedical field, including drug delivery and tissue engineering, due to its tunable porous morphology and convenient surface bio-functionalization [14]. PSi appears to be a promising biomaterial for tissue engineering as it is nontoxic, biocompatible and bioresorbable under physiological conditions, and dissolves progressively into nontoxic silicic acid. Soluble silicic acid is the biologically active form of silicon, and is mainly absorbed from silicic acid-containing drinking water, beverages made from phytoliths-containing plants, and some food (mainly banana and green bean) [15,16]. Silicon plays an important role in connective tissue, such as bone and cartilage, for the formation of the organic matrix (collagen and glycosaminoglycan), and participates in the biochemistry of osteogenic cells, highlighting its metabolic role in connective tissue [17]. PSi is formed by the electrochemical etching of pure crystalline silicon in a hydrofluoric acid (HF) solution. It has several advantages in comparison with existing materials. Pore dimensions can be precisely controlled in a large range: from micropores (<2 nm) to mesopores (2–50 nm) and up to macropores (>50 nm up to several microns). Another useful feature of pSi is its high-surface area (400–1000 m^2^/g), which, coupled with the ability to control pore sizes, allows the load of a range of bioactive species [18]. Its degradation kinetics can be controlled oxidation rate and surface functionalization. In addition, oxidation and subsequent silanization can alter surface properties and favor cell attachment, depending upon the functional groups of the silane [19,20,21,22]. Coupling the auspicious capacities of human adult mesenchymal stem cells with the unique properties of pSi substrates provides a promising approach for therapeutic application in regenerative medicine. In a previous study, we had observed that optimal DPSC adhesion and proliferation was obtained for porous silicon substrates of 30 to 40 nm pore diameters with amine functionalized surfaces [9]. We aimed to evaluate pSi platforms associated or not with DPSC in a pre-clinical model of bone reconstruction. For this, we used a rat-tail model targeting bone regeneration studies, with the possibility of multiple testing within the same experimental animal, following ethical recommendations [23]. Critical-sized defects were established to follow bone formation in response to the various treatments, and a specific design of four sites on the same animal was elaborated. We created a critical-sized defect in rat tail vertebrae and followed bone healing with (and without) DPSC, and with (and without) pSi.

## 2. Materials and Methods

### 2.1. Porous Silicon Scaffolds

Porous silicon platforms with pore diameter of 30–40 nm were created from highly type boron-doped crystalline silicon wafers from Siltronix (Siltronix, Archamps, France) with 0.0008 to 0.0012 Ωcm resistivity. Wafers were etched in a custom-made Teflon cell at a constant current density of 300 mA/cm^2^ for 20 min in a hydrofluoric acid (HF) solution in ethanol (3:1 HF/ethanol solution, volume ratio). Etched layers were detached by applying a 5 mA/cm^2^ current for 30 s, then sonicated for 1 min in an ultrasonic bath. The obtained pSi particles were dried before being thermally oxidized at 400 °C for 1 h. Particles were then resuspended in 70% ethanol and silanized with APTES (aminopropyltriethoxysilane, Sigma Aldrich, Saint Louis, MO, USA). The obtained aminated particles were stored in ethanol until experiments.

The pSi pore diameters were evaluated by environmental scanning electron microscopy (SEM, Analytic FEI Quanta FEG 200, FEI, Hillsboro, OR, USA) with an acceleration voltage of 20.00 kV in a pressure of 0.5. Torr. ImageJ^®^ software (FiJi version, GNU General Public License v3) was used to measure the mean pore diameter.

### 2.2. Xenogenic Bone Substitute (XBS)

To provide an efficient comparison to pSi particles, Xenogenic Bone Substitute (XBS) particles already available on the market were used as the control. We used Bio-Oss^®^ products that are commercially available bone granules of bovine origin (Bio-Oss^®^, Geistlich Pharma AG, Wolhusen, Switzerland), in which organic components have been removed without alteration of the microstructure and mineral part. Thus, these XBS particles harbor a structure and composition that resembles mineralized human bone.

### 2.3. Human Dental Pulp Stem Cells Collection

Human mesenchymal stem cells from dental pulp (DPSC) were recovered from impacted third molars, which were extracted for medical reasons from adult subjects aged 16 to 30. Each subject provided informed written consent following a protocol approved by the local ethical committee (Montpellier hospital, France). The recovered teeth were opened in a laminar flow hood using a diamond disk, and dental pulps were gently removed and digested in a pulp enzymatic digestive solution of 3 mg/mL collagenase type I and 4 mg/mL dispase (BD Biosciences, supplemented with 10% fetal bovine serum (Invitrogen, Carlsbad, CA, USA). Non-adherent cells were removed by a change of medium 24 h after cell seeding. After 1 week, subconfluent cells were collected, then controlled for stem cell markers by flow cytometry and seeded on pSi scaffolds. 

### 2.4. Fluorescence Microscopy

DPSC attachment was monitored by fluorescence microscopy. Cells were seeded onto the surface of sterilized pSi particles at a cell density of 5 × 10^4^ cells/mL. Cells were incubated for 24 h at 37 °C with 5% CO_2_ in a humidified incubator, in ultra-low adherence 24 well plates (Corning, NY, USA) that inhibit cell attachment on the tissue culture plate, allowing DPSC to attach only to the pSi particles. After the incubation time, the cells were fixed in 2.5% glutaraldehyde and stained for actin cytoskeleton and nuclei. Cells were permeabilized with 0.5% Triton X-100 in PBS at 4 °C for 15 min, incubated for 1 h with TRITC-labeled phalloidin (1:200) at 37 °C in the dark, then with Hoechst 33342 (Life Technologies, Carlsbad, CA, USA) for 10 min at room temperature, and washed twice with deionized water. Samples were observed under fluorescence microscopy (Nikon TE2000, Nikon Instruments, Amsterdam, The Netherlands) at an excitation wavelength of 360 nm for Hoechst staining and 540 nm for phalloidin labeling.

### 2.5. Scanning Electron Microscopy

Cells were cultured on pSi and XBS particles for 24 h under normal conditions. After incubation, cells were washed with PBS and fixed with 2.5% glutaraldehyde for 1 h at room temperature. Samples were dehydrated in graded ethanol solutions from 50 to 100% and in hexamethyldisilazane (HMDS, Ted Pella, Redding, CA, USA), then sputter-coated with platinum. Scanning electron microscopy (SEM) was performed on an Analytic FEI Quanta FEG 200 microscope (FEI, Hillsboro, OR, USA) with an acceleration voltage of 15 kV. 

### 2.6. Animal Experiments

The study was approved by the ethical committee for animal welfare of Montpellier University (referral number 1083 16 June 2014).

Two groups (5 rats per group) of male Wistar rats ((Crl:(Wi)Br) from Charles River France), with weights ranging from 380 to 450 gr, were used for an adequate vertebrae size. All animals were kept in light controlled, air-conditioned rooms and fed ad libitum. The groups of rats were implanted, respectively, for one- and two months periods. In each rat, four vertebrae were used for biomaterial implantation. Each defect was implanted with either randomized pSi particles alone, pSi particles carrying DPSC, XBS particles, or negative control (defect left blank). Animals were anesthetized with an intraperitoneal injection of ketamine and xylazine (Alcyon, Pau, France) (40 and 9 mg/kg, respectively). As previously described, a dorsal incision was made after proper disinfection [23]. The skin and muscles were retracted and the vertebrae were exposed. Critical-sized bone defects of approximately 20 mm^3^ (3 × 3 mm) were realized in the exposed surface of 4 vertebrae, and defects were manually filled with each biomaterial. After implantation, the muscles were repositioned over the defects and sutured together with resorbable sutures (Vicryl 4/0, Ethicon, Issy les Moulineaux, France). Then, the skin was tightly sutured with resorbable sutures (Vicryl 3/0, Ethicon, Issy les Moulineaux, France). Following surgery, Buprenorphine SR-LAB (1 mg/mL), (Wildlife Pharmaceuticals, Windsor, CO, USA) at a dose of 1.2 mg/kg was used for systemic relief and to provide 72 h analgesia. Rats were kept in individual cages and the wound healing was controlled daily for the first week and twice per week during the following healing periods. Every second day the tails were disinfected using povidone-iodine solution (Betadine, Mundipharma, Paris, France). At the end of the different long experimental periods, the rats were sacrificed by intraperitoneal injection of Pentothal (Alcyon, Pau, France) with a suitable dosage (200 mg/kg). The tails were harvested. The soft tissues were carefully removed mechanically and the samples were fixed in 5% formaldehyde solution in 4 °C for 24 h. 

### 2.7. MicroCT Analysis

The morphology of the reconstructed vertebrae was assessed using a micro-CT system (SKYSCAN 1172 X-ray Microtomograph, Microphotonics Inc., Allentown, PA, USA) and a 3D reconstruction software (Avizo, FEI company, Hillsboro, USA). Implanted samples were planned to be scanned at 360° rotation at 0.7 degree intervals. Measurements were made on the Region of Interest (ROI) × 1.5 mm Tissue Volume (TV) on the computer-reconstructed 3D samples. These measurements were performed before the other analyses. Three-dimensional reconstructions were permitted to evaluate bone density into the defect in function with the biomaterials used and in function with the experimentation time. To approach the reconstructed bone volume within the drilled area, a ball of the diameter of the drilling form was built into the Avizo software. Then, an extra volume containing the region of interest was extracted. After segmentation, the calculation of bone volume was realized, as well as the total volume of the ball. The bone mineral densities (BMD) were used for comparison in this study. The osteogenic properties of the biomaterial were evaluated by the comparison of the bone mineral density (BMD), expressed in g/cm^3^. Hydroxyapatite pellets of 0.25 and 0.75 g/cm^3^ were used as references.

### 2.8. Histology and Immunohistochemistry

The specimens were fixed overnight with paraformaldehyde solution 4% in phosphate-buffered saline PH 7.4 and were decalcified in 10% EDTA supplemented with 1% of paraformadehyde during 10 days in microwave (Milestone Histos 5 Rapid, Tissue processor, Birmingham, Italy). Samples were washed in water before processing for sucrose-gelatine impregnation and freezing in isopenthane at −80 °C. Then, slices were cut (55 µm thick) along the coronal plane through the center of each defect with cryostat (Leica 3050S, Nussloch, Germany). For histology, the sections were stained with hematoxylin and eosine. Finally, after three washes in distilled water, the samples were passed through a graded sequence of alcohols (95°, 75°, and 95°) and mounted on slide with the mounting medium. For immunostaining, 50 µm floating sections in culture dishes were blocked in PBS supplemented with 10% goat serum and 0.3% Triton for 30 min and incubated with primaries antibodies osteocalcin (mouse Abcam, ab13418) and HuNu (mouse Anti-Human Nuclear Antigen antibody -235-1, ab191181) during 3 days at 4 °C with gentle agitation on different slides. Sections were washed 3 times in PBS during 3 h, and were incubated with secondary antibodies overnight at 4 °C (goat anti-mouse coupled to Alexa Fluor 488 (Molecular probes) and DAPI (Sigma)) and washed again 3 times before being mounted in Dako Fluorescent anti-fading Mounting Medium. Cryosections were imaged on a Confocal Leica SP5 Fluorescent. All images were analyzed with ImageJ software. For immunohistochemistry, positive staining was quantified after the threshold and results were normalized, according to cell number per image (counted with DAPI staining), in order to be expressed as a percentage of positive cells.

### 2.9. Statistical Analysis

Data was reported as the mean ± standard error. All post-acquisition image processing was conducted with ImageJ software (FiJi version, GNU General Public License v3). Statistical analysis was performed using SigmaStat (Version 3.0, Inpixon HQ, CA, USA). All data were checked for normalization. When applicable, data were analyzed using parametric One-Way Analysis of Variance (ANOVA). When a normality test failed, data were analyzed with Kruskal–Wallis One Way Analysis of Variance on Ranks. For all tests, significance was determined to be *p* < 0.05.

## 3. Results

Dental Pulp Stem Cells (DPSC) adhesion on pSi scaffolds was evaluated with epifluorescence microscopy after cytoskeleton and nucleus staining (TRITC-labeled phalloidin and Hoechst 33342). The pSi scaffolds were prepared as micron-sized porous pSi particles, as previously described [9]. Effective cell adherence on pSi particles was observed after 24 h of incubation (Figure 1), with cell spreading on pSi particle surfaces. Cells were hardly attaching onto XBS particles, with the few attached cells presenting an unorganized cytoskeleton (Figure 1). 

The particles’ shape and topography (pSi and XBS) were investigated with SEM (Figure 2). PSi particles featured a uniform rectangular shape, whereas XBS granules were non-uniform with a surface similar to bone structure. Mean size of pSi particles was 153 ± 57 µm. XBS granules mean size was 682 ± 78 µm. Both pSi and XBS displayed a porous top surface; pSi substrates presented a uniform porosity in comparison to XBS. Surface characterization showed an expected mean pore diameter of 33 ± 5 nm for pSi, and pores roughly ranging from 5 to 10 µm for XBS (Figure 2B). Cells appeared elongated and well spread with the formation of protrusions out of the cell membrane on pSi, while no cells were visible on the XBS substrate (Figure 2).

Once correct cell adhesion was confirmed in vitro, in vivo experiments were designed, taking in account that cells did not attach properly on XBS. Thus, bone healing was monitored in the four following cases: (i) defect left blank (negative control), (ii) defect filled with pSi particles, (iii) defect filled with pSi particles carrying the allogenic human stem cells, and (iv) defect filled with XBS granules (positive control). The surgical procedure is illustrated in Figure 3. It includes dorsal incision, skin and muscle retraction to expose the operative site (Figure 3A,B). Figure 3C shows the corresponding radiograph slide of a critical size defect of 3 × 3 mm in diameter into the vertebrae. During the whole animal experimentations, no infection of the operative site was observed. Rat behavior did not change during experiments and post-surgical care period. Specific attention was provided to signs of inflammation, especially linked to the allogenic cells graft. No signs of inflammation linked to cell graft were observed. Filling materials were stable in the defects after immediate positioning, as particles were contained in a four-wall bony defect. No loss of materials and no infection of the operative sites were observed after the different periods of healing (Figure 3E,F).

After 1 and 2 months, vertebrae were analyzed by µCT to assess mineralization and bone tissue formation. After 1 month, bone mineral density (BMD) was significantly higher in vertebrae filled with pSi particles (with or without DPSC), compared to control vertebrae (defect left blank) (Figure 4). BMD was significantly higher for defects filled with XBS particles as they are dense mineral particles. No statistical differences could be observed with pSi with or without cells. After 2 months, BMD was still significantly lower for control defects and significantly higher for defects filled with XBS particles, compared to pSi particles. Interestingly, an increased BMD, similar to the native vertebrae bone density, was observed when pSi carried DPSC (Figure 4).

After the period of experiment, vertebrae were fixed, decalcified, and cut for histological examination after hematoxylin-eosin staining (Figure 5). No evident bone formation was observed when the defects were left empty, after either 1 or 2 months. In contrast, new bone formation was clearly visible in all the cases where defects were filled with particles. The bone formation mainly started from the border of the defect, at the interface between filling materials and native bone (represented with green dash lines in Figure 5). In comparison with XBS granules, a denser bone matrix was generated around pSi particles (with or without DPSC). When comparing bone healing at 1 and 2 months, bone formation increased towards the center of the defect. Considering all the samples, bone matrix was substantially more abundant in defects filled with pSi particles (with and without DPSC) than in XBS sites. PSi particles could hardly be seen after 1 or 2 months: only a few of them remained visible after 1 month. On the contrary, XBS particles were clearly visible even after 2 months (lacuna in the decalcified samples, visible in Figure 5D). Active new bone formation was evidenced by the appearance of osteoid and typical lamellar bone morphology, similar to native bone with osteocytes and with newly formed bone lined by osteoblasts. In addition, bone matrix appeared denser in defects where DPSC had been grafted, compared to defects filled with pSi particles alone, especially in the samples observed after 1 month, indicating an increase in early bone regeneration (Figure 5). Denser bone matrix was observed after 2 months within the pSi particles alone group with less void space, confirming the efficiency of pSi particles for bone regeneration, but with delayed effect in the absence of DPSC (Figure 5F).

Engineered bone tissue was explored by immunofluorescence in order to evaluate the cellular content of the regenerated tissue at the early stage of healing (osteocalcin activity within the first month of healing). Fluorescence images from confocal microscopy are presented in Figure 6. Active bone formation was enlightened through osteocalcin (OCN) staining. Osteocalcin activity was quantified as a percentage of positive cells (number of cells expressing OCN compared to total number of cells labeled by DAPI staining). After 1 month, OCN activity was hardly visible with the negative control (defect left blank) but was important for all the biomaterials used, indicating an active bone formation process. Significant differences could be observed between XBS and pSi particles: OCN activity was significantly more important with the use of pSi particles combined with DPSC, compared to XBS or to pSi particles alone (Figure 6G). However, no difference was found between pSi particles alone and XBS particles. As DPSC seemed to have a significant impact on the bone formation, the persistency of the grafted human DPSC was investigated after 1 month with immunostaining, using the anti-Human Nuclear (HuNu) antibody. Hence, if any DPSC of human origin remained in the rat vertebrae, they would be stained by the HuNu antibody. Surprisingly, this staining revealed the absence of human cells in the surgical site: all the cells in immunohistological slices were rat cells that did not express the HuNu antigen. To exclude any risk of false negative results, the efficacy of the HuNu antibody labeling was controlled on human cells (Figure 6H–J).

## 4. Discussion

The aim of this work was to investigate the in vivo potential of pSi particles as resorbable bone substitutes, as well as the impact of combination with DPSC. PSi surfaces were chemically functionalized with amino groups and pore diameters were tuned to optimized cell adhesion and proliferation (30–40 nm), in accordance to previous works following DPSC adhesion on pSi chips [9,24]. Consistently, in vitro experiments confirmed efficient adhesion and spreading of DPSC on pSi particles. As XBS granules did not support DPSC growth, experiments were designed without any stem cells on XBS particles, which corresponds to its normal clinical use as a filling bone substitute. PSi particles were generated by electrochemical etching of crystalline silicon, then fractured in an ultrasonic bath to reach the appropriate size (100–200 µm) [25,26,27]. Particles size was determined in order to permit cell adhesion and spreading, and to favor vascularization in the empty spaces between particles [28]. The produced pSi particles, which were two to three times smaller than XBS particles, were also shown to properly fill bone defects in vertebrae. The analysis of bone mineral density (BMD) after 1 and 2 months highlighted the positive impact of pSi on bone regeneration. BMD increased over time, and even continued to increase after pSi particles degradation into silicic acid (Figure 4). This observation is consistent with previous studies indicating that silicic acid stimulated osteogenic differentiation and bone mineralization [10,29,30]. This increase is also consistent with the appearance of new bone replacing the grafted pSi particles (Figure 5). BMD was maximal for defects filled with XBS particles (even higher than native bone), but without an increase over time. This is related to the nature of the material, which is composed of dense Calcium-Phosphate granules that did not dissolve over time with limited bone matrix embedding compared to pSi particles. Furthermore, after 1 month, no DPSC could be recovered in samples where they had been grafted (Figure 6). However, OCN activity after 1 month was significantly higher when DPSC were grafted. For the DPSC-pSi group, the significant increase of OCN activity after 1 month illustrates DPSC paracrine signaling: DPSC are mesenchymal stromal cells with proliferation-differentiation capacities and paracrine activities that are relevant properties for regenerative therapies. Grafted cells proliferation-differentiation capacities are supposed to contribute by regenerating damaged bone, whereas paracrine activities regulate the cellular response to injury. Paracrine properties are due to DPSC soluble factors release. These factors influence the microenvironment by stimulating resident cells (osteoblasts, fibroblasts, and vascular cells) and by modulating the host’s immune response. They may act directly on injured cell mechanisms, or indirectly, with the secretion of mediators by neighboring cells. Various properties of mesenchymal stem cells have been described, including attenuation of tissue injury, inhibition of fibrotic remodeling and apoptosis, angiogenesis promotion, stem cell recruitment and proliferation, and oxidative stress reduction [31,32]. Finally, after 1 month, DPSC increased bone matrix formation around pSi particles (Figure 6) with no significant difference in BMD (Figure 4). After 2 months, the difference in terms of bone matrix formation was less obvious, but a significantly higher BMD was observed where DPSC had been grafted (Figure 4), indicating that the presence of DPSC accelerates the bone healing process, even though they were not proliferating for long term in vivo. DPSC showed a durable effect on bone healing that could be objectivated even after cells disappeared from the grafted site, as described in the concept of stem cell therapy without the cells [33].

PSi materials also favored bone regeneration, influencing both mineralization (calcium phosphate nucleation) and cellular activity through silicic acid release, as previously described [34,35,36,37,38].

In this study, we used an animal model to evaluate bone regeneration for dental and maxillofacial surgery, as rat caudal vertebrae are similar to human jaw bones [23]. This model allows the use of four surgical sites in the same animal, thus limiting the number of required animals and the bias of animal-related variations. We were able to test three conditions (plus negative control) per animal, and compare the results for each subject. The use of pSi particles was as pr”ctic’l as XBS particles, and led to a bone regenerative process following pSi resorption. The overall mineral density was higher with XBS particles filling, but extracellular matrix was more abundant around pSi than around XBS, leading to the conclusion that pSi had higher biological activity. When DPSC were grafted with pSi, rat osteoblasts were significantly more active, although human cells had disappeared. This higher activity became clinically relevant after 2 months, with higher BMD for pSi-DPSC group compared to pSi group.

## 5. Conclusions

Techniques using bone substitutes alone have limited efficiencies. Therefore, bone tissue engineering, using biomaterials and stem cells, is often considered as future therapeutics to rebuild defects that could not heal spontaneously. We have investigated the potential for in vivo bone regeneration of a combination of pSi scaffolds and DPSC. Histological analyses revealed collagen network formation, and immunohistochemical assays confirmed the increased expression of bone markers. Surprisingly, no grafted cells remained in the regenerated area after one month of healing, even though the grafting of DPSC clearly increased bone regeneration for both bone marker expression and overall bone formation objectivated with µCT. Our results provide in vivo proof of the efficiency of pSi with DPSC, with greater results than those obtained with simple material grafts, highlighting the paracrine role of grafted cells to recruit and stimulate endogenous cells. 

Thus, we successfully evaluated the efficacy of pSi particles for bone regeneration in critical size defects. The resorbable biomaterial contributed to increase bone matrix formation, with a kinetic consistent with particles resorption. Matrix mineralization started during the first month of healing and continued for at least 2 months. The addition of mesenchymal stem cells (DPSC) carried by pSi particles enhanced cell biological activity and accelerated the overall process of bone regeneration and repair by stimulating the resident rat cells through paracrine actions. These results open important perspectives towards the development of “stem cell therapy without cells” as a promising approach for bone tissue reconstruction.

## Figures and Tables

**Figure 1 bioengineering-10-00852-f001:**
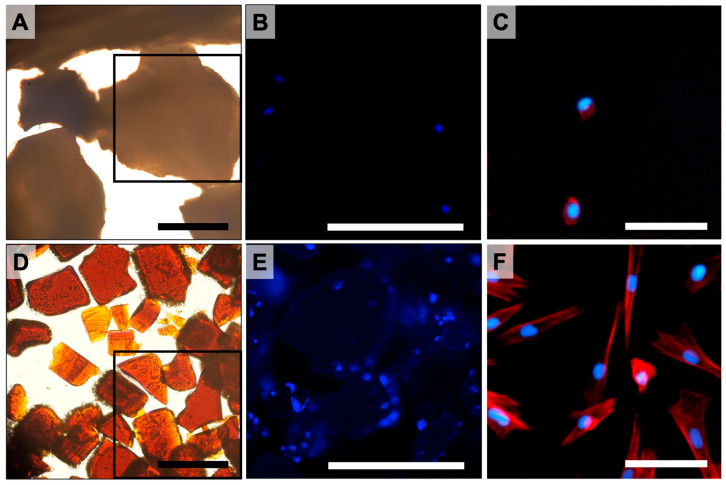
DPSC incubated on XBS and pSi particles. (**A**–**C**): XBS particles. (**D**–**F**): pSi particles. (**A**,**D**): brightfield microscopy of XBS and pSi particles at magnification ×20 (scale bar = 100 µm). (**B**,**E**): epifluorescence microscopy corresponding to the indicated squares in picture (**A**) and (**D**), respectively (cell nuclei stained in blue). (**C**,**F**): epifluorescence microscopy at higher magnification showing cytoskeleton organization (actin stained in red). (**A**,**B**,**D**,**E**): Scale Bar = 150 µm. (**C**,**F**): Scale Bar = 40 µm.

**Figure 2 bioengineering-10-00852-f002:**
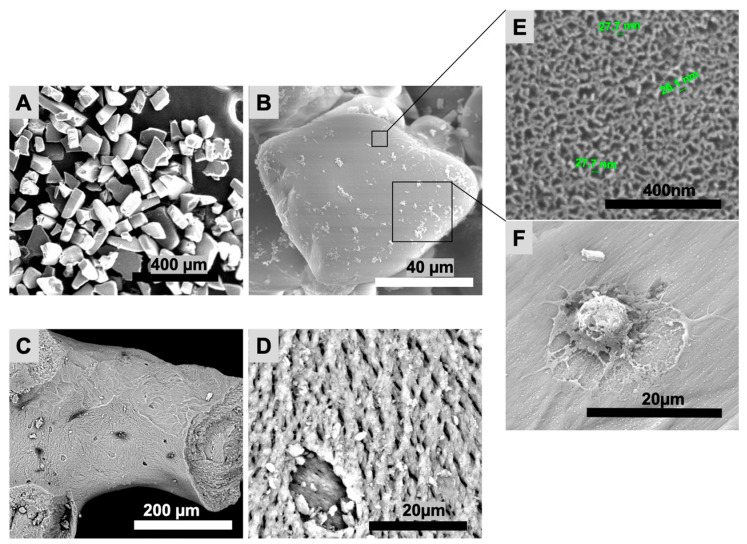
Scanning electron microscopy (SEM) of pSi and XBS particles. (**A**,**B**): pSi particles. (**C**,**D**): XBS particles. (**E**): pSi surface at high magnification, showing the nanoporous structure. (**F**): DPSC on pSi with formation of protrusions out of the cell membrane.

**Figure 3 bioengineering-10-00852-f003:**
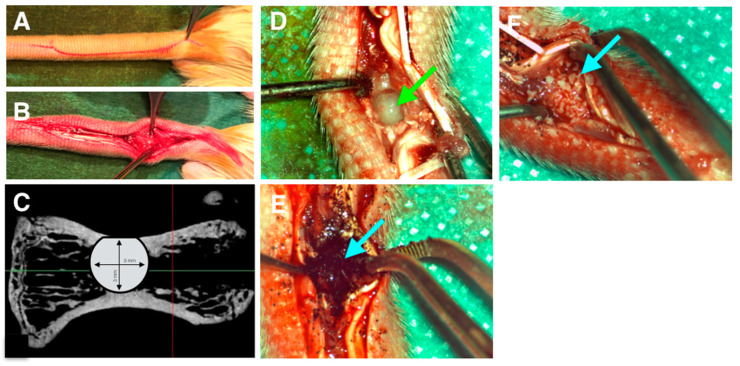
Surgical procedure. (**A**): skin incision. (**B**): skin and muscles retraction. (**C**): critical size defect plan on retro-alveolar radiograph. (**D**): calibrated bone defect on vertebrae. Green arrow indicates empty defect. (**E**): defect filled by porous silicon particles. (**F**): defect filled by XBS. Blue arrow indicates filled defect.

**Figure 4 bioengineering-10-00852-f004:**
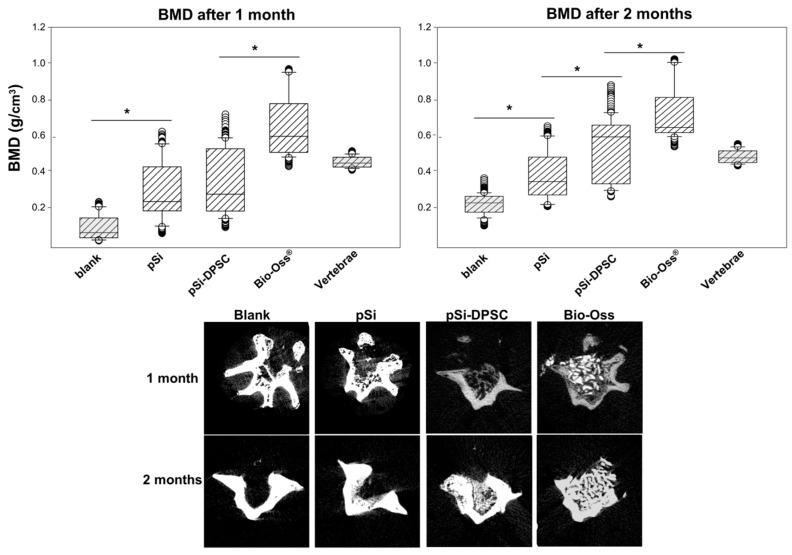
Analyses by µCT after 1 and 2 months. Upper part: Bone mineral density comparisons between every test and control at 1 and 2 month of healing. * represents significant differences. Lower part: µCT slices of defect filled by pSi particles, pSi particles with DPSC, XBS particles and negative control, after 1 and 2 months.

**Figure 5 bioengineering-10-00852-f005:**
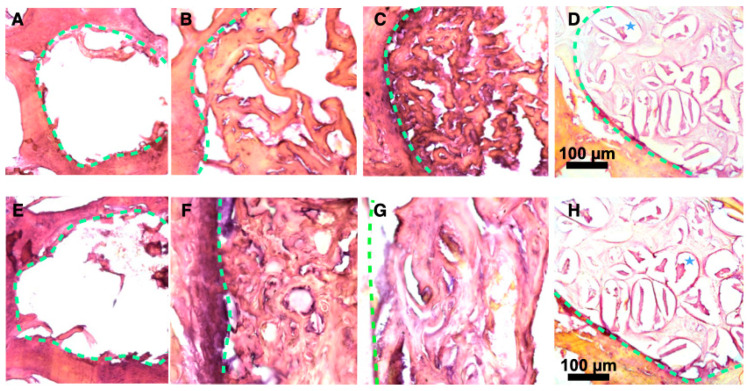
Histological slices after 1 and 2 months ((**A**–**D**) and (**E**–**H**), respectively). Slices were stained with Hematoxylin-Eosin. (**A**,**E**) Blank defect. (**B**,**F**) pSi particles. (**C**,**G**) pSi particles with DPSC. (**D**,**H**) XBS granules. PSi particles are not visible after 1 or 2 months. Dash lines represent junctions between original bone and new bone. Blues stars represent demineralized XBS particles.

**Figure 6 bioengineering-10-00852-f006:**
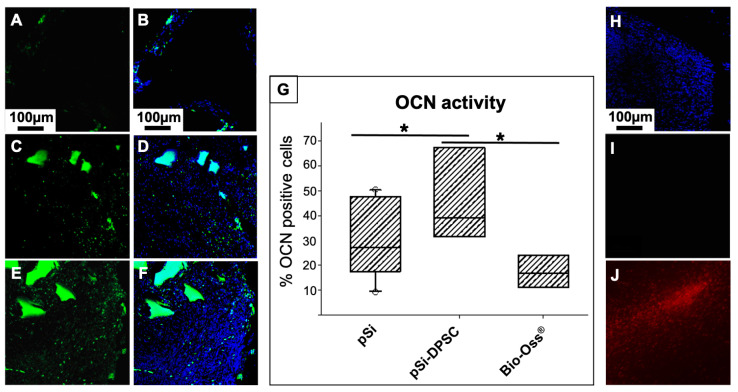
Osteocalcin (OCN) activity: Fluorescence images of bone defects after 1 month of healing, filled with XBS particles (**A**,**B**), pSi particles alone (**C**,**D**), and pSi particles carrying DPSC (**E**,**F**). OCN is stained in green (**A**,**C**,**E**), and corresponding merged image with cell nuclei stained in blue are presented in (**B**,**D**,**F**), respectively. The box plot graph (**G**) represents OCN activity, measured as a percentage of positive cells. * represents significant differences. (**H**–**J**) HuNu staining on slices from vertebrae after 1 month of healing to highlight the presence or absence of human DPSC: (**H**) Nuclear staining (DAPI) in blue showing the presence of cells, (**I**) HuNu labelling in red (no signal visible) enlightening the absence of human cells, (**J**) Positive control of HuNu labelling efficiency (red) on human cells confirming that the absence of signal corresponds to an absence of human cells.

## Data Availability

The data presented in this study are available on request from the corresponding author. The raw/processed data required to reproduce these findings cannot be shared at this time due to technical limitations.

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
