# Peer review of "Allogenic Stem Cells Carried by Porous Silicon Scaffolds for Active Bone Regeneration In Vivo"

_bioengineering, 2023, doi:10.3390/bioengineering10070852_

Round 1
Reviewer 1 Report
The work entitled “Allogenic Stem Cells carried by Porous Silicon Scaffolds for Active Bone Regeneration in vivo” presents the development of porous microparticles that can be efficiently used as stem cell carriers for the restoration of bone defects. The results from in vitro and in vivo experiments are indeed relevant to the fields of tissue engineering and stem cell-based therapies, as they provide an alternative approach to fabricating highly bioactive scaffolds, with comparable or even better performance than that of their currently commercial counterparts. To improve the quality of the manuscript, please address the following:
1. Figure 1: visual inspection of Figures 1D and 1E indicates that the staining included cells that were not attached to the particles, but to the bottom of the well. This can be easily evidenced by looking, for example, at the area on the upper right of each picture: while the brightfield image (Figure 1D) shows an empty (particle-free) area, the fluorescence image shows the same area full of cell nuclei (Figure 1E). In fact, several particle-free zones shown in the brightfield picture are covered by cells in the corresponding fluorescence image. This is extremely misleading and should be corrected because it creates a visual bias.
2. The authors claimed that “In addition, bone matrix appeared denser in defects where DPSC have been grafted, compared to defect filled with pSi particles alone, especially in the samples observed after 1 month, indicating an increase in early bone regeneration (Figure 5).” This statement seems inaccurate, since comparison of Figures 5F and 5G renders the pSi group (Figure 5F) with better results after 2 months (less void space). Please, clarify this in the revised version of the manuscript.
3. Please, carefully revise the manuscript to correct for some typos. For example, one recurrent typo is “plateform”, which should be written as “platform”.
4. Following the previous comment, a more adequate definition of the studied material would be “particles”.
Please, carefully revise the manuscript to correct for some typos.
Author Response
ANSWERS TO REVIEWER 1
We thank the reviewer for the careful and precise evaluation of the submitted work. We have added corrections to the manuscript according to the remarks and recommendations (corrections in red in the manuscript).
Please find below some answers to the specific comments :
- Figure 1: visual inspection of Figures 1D and 1E indicates that the staining included cells that were not attached to the particles, but to the bottom of the well. This can be easily evidenced by looking, for example, at the area on the upper right of each picture: while the brightfield image (Figure 1D) shows an empty (particle-free) area, the fluorescence image shows the same area full of cell nuclei (Figure 1E). In fact, several particle-free zones shown in the brightfield picture are covered by cells in the corresponding fluorescence image. This is extremely misleading and should be corrected because it creates a visual bias.
We have modified Figure 1, especially 1B and 1E, to show fluorescent images focusing on area where ony particles could be seen (and not the bottom of the well)
- The authors claimed that “In addition, bone matrix appeared denser in defects where DPSC have been grafted, compared to defect filled with pSi particles alone, especially in the samples observed after 1 month, indicating an increase in early bone regeneration (Figure 5).” This statement seems inaccurate, since comparison of Figures 5F and 5G renders the pSi group (Figure 5F) with better results after 2 months (less void space). Please, clarify this in the revised version of the manuscript.
Indeed, after 2 months, the histological results show denser bone matrix in the pSi group without cells, reaching the matrix density observed in the pSi-DPSC group after 1 month. This observation illustrates the difference of bone regeneration kinetics, and the increase in early bone regeneration when DPSC are added. We have added some clarifications in the manuscript (in red in the manuscript)
- Please, carefully revise the manuscript to correct for some typos. For example, one recurrent typo is “plateform”, which should be written as “platform”.
We have revised the manuscript to corrects typos. i.e « plateform » has been replaced by « platform », « BioOss » has been replaced by « Xenogenic Bone Substitute » or « XBS ». Corrections are written in red in the revised manuscript.
- Following the previous comment, a more adequate definition of the studied material would be “particles”.
We fully agree that the studied material should be described as particles. We have used 2 times the term « platform » to illustrate that the particles could be used as cell carriers.

Reviewer 2 Report
The manuscript describes a bone regeneration of porous silicon scaffolds with dental pulp stem cells compared to a commercialized bone substitute product in a rat-tail bone defect model. The manuscript is well organized and performed. But, to make it better, further improvements should be made by the following minor revision.
1. Figure 5: Trichrome staining is highly required to determine whether it is scaffold or regenerated bone.
2. Figure 6: Merged images of cell nuclei and OCN should be required. Also, a detailed description for Figure 6H-J is needed. This is difficult to understand with only the given information and description.
Minor editing of English language required.
Author Response
ANSWERS TO REVIEWER 2
We thank the reviewer for the careful and precise evaluation of the submitted work. We have added corrections to the manuscript according to the remarks and recommendations (corrections in red in the manuscript).
Please find below some answers to the specific comments :
- Figure 5: Trichrome staining is highly required to determine whether it is scaffold or regenerated bone.
This remark is very relevant and we agree that Trichrome staining would have been of high interest. However, we had to decalcify sufficiently to properly cut the samples with the remaining particles. Therefore, we faced difficulties to obtain efficient trichrome staining, and we only managed to work with HE staining.
- Figure 6: Merged images of cell nuclei and OCN should be required. Also, a detailed description for Figure 6H-J is needed. This is difficult to understand with only the given information and description.
We have modified Figure 6, to include merged images of cell nuclei and OCN. We have added descriptions of HuNu labelling (Fig 6H-J) in the main text and in the figure legend (corrections and additions in red)
